# Impact of Baseline Clinical Variables on SGLT2i’s Antiproteinuric Effect in Diabetic Kidney Disease

**DOI:** 10.3390/life13041061

**Published:** 2023-04-21

**Authors:** Irene Capelli, Danilo Ribichini, Michele Provenzano, Daniele Vetrano, Valeria Aiello, Giuseppe Cianciolo, Valentina Vicennati, Alessandro Tomassetti, Ginevra Moschione, Sabrina Berti, Uberto Pagotto, Gaetano La Manna

**Affiliations:** 1Nephrology, Dialysis and Renal Transplant Unit, IRCCS Azienda Ospedaliero-Universitaria di Bologna, 40138 Bologna, Italy; 2Department of Medical and Surgical Sciences (DIMEC), Alma Mater Studiorum University of Bologna, 40138 Bologna, Italy; 3Division of Endocrinology and Diabetes Prevention and Care, IRCCS Azienda Ospedaliero-Universitaria di Bologna, 40138 Bologna, Italy

**Keywords:** sodium-glucose cotransporter 2, SGLT2 inhibitors, proteinuria, diabetes, chronic kidney disease, diabetic kidney disease

## Abstract

Introduction: Proteinuria is a major risk factor for the progression of chronic kidney disease (CKD). Sodium-glucose cotransporter 2 inhibitors (SGLT2i) demonstrated a nephroprotective and antiproteinuric effect in people with type 2 diabetes (T2DM) and proteinuric CKD. We conducted a retrospective study to evaluate clinical and laboratory variables that can help predict proteinuria reduction with SGLT2i therapy. Materials and methods: Patients affected by T2DM and CKD who started any SGLT2i were included in the study. Patients were stratified into two subgroups, Responder (R) and non-Responder (nR), based upon the response to the therapy with SGLT2i, namely the reduction in a 24 h urine proteins test (uProt) of ≥30% from baseline levels. The aim of the study is to analyse differences in baseline characteristics between the two groups and to investigate the relationship between them and the proteinuria reduction. A Kruskal–Wallis test, unpaired t-test and Chi^2^ test were used to test the difference in means and the percentage (%) between the two groups. Linear and logistic regressions were utilized to analyse the relationship between proteinuria reduction and basal characteristics. Results: A total of 58 patients were enrolled in the study: 32 patients (55.1%) were in the R group and 26 patients (44.9%) in the nR group. R’s patients had a significant higher uProt at baseline (1393 vs. 449 mg/24 h, *p* = 0.010). There was a significant correlation between baseline uProt and proteinuria reduction with SGLT2i in both univariate (β = −0.43, CI −0.55 to −031; *p* < 0.001) and multivariate analyses (β = −0.46, CI −0.57 to −0.35, *p* < 0.001). In the multivariate analysis, there was a significant positive correlation between the estimated glomerular filtration rate (eGFR) and proteinuria reduction (β = −17, CI −31 to −3.3, *p* = 0.016) and a significant negative correlation with body mass index (BMI) (β = 81, CI 13 to 50, *p* = 0.021). The multivariate logistic regressions show a positive correlation of being in the R group with diabetic retinopathy at baseline (Odds Ratio (OR) 3.65, CI 0.97 to 13.58, *p* = 0.054), while the presence of cardiovascular disease (CVD) at baseline is associated with being in the nR group (OR 0.34, CI 0.09 to 1.22, *p* = 0.1), even if these statements did not reach statistical significance. Conclusions: In this real-life experience, following the administration of SGLT2i, a reduction of more than 30% in proteinuria was observed in more than half of the patients, and these patients had a significantly higher baseline proteinuria value. Variables such as eGFR and BMI are variables that, considered in conjunction with proteinuria, can help predict treatment response before therapy initiation. Different phenotypes of diabetic kidney disease may have an impact on the antiproteinuric response.

## 1. Introduction

Proteinuria is not only a marker of kidney impairment, but also a major risk factor for cardiovascular disease (CVD), the progression of chronic kidney disease (CKD) [1,2] and acute kidney injury [3]. Several studies have demonstrated that protein excretion in urine can directly cause damage to podocytes, leading to their stimulation and production of transforming growth factor beta (TGF-β). This, in turn, induces the differentiation of mesangial cells into myofibroblasts [4,5]. On the other hand, protein overload in the tubules triggers the release of cytokines, chemokines, growth factors and vasoactive molecules from the tubular cells. This results in an abnormal build-up of inflammatory cells, extracellular matrix collagen, fibronectin and other components in the interstitial space, which can lead to interstitial fibrosis. The tissue injury causes renal cells to generate reactive oxygen species and undergo endoplasmic reticulum stress response. This leads to the modification of membrane lipids, proteins and DNA due to oxidative stress, which triggers cell-death responses, tissue inflammation and the recruitment of macrophages and lymphocytes. These events further contribute to the ongoing inflammatory process [6,7].

Proteinuria is a common feature of diabetic kidney disease (DKD), being present despite proper treatment, in a striking proportion (around 70%) of patients referred to nephrologists [8]. It is believed that proteinuria is a late manifestation of glomerular hyperfiltration, which is the main pathological mechanism implicated in the development of kidney damage in diabetic patients [9]. In DKD, the extent of protein loss is closely associated with both renal and patient survival, and nephrotic proteinuria is a significant risk factor for accelerated kidney damage, leading to premature end-stage kidney disease (ESKD), cardiovascular disease and mortality [10,11,12].

Reducing proteinuria is associated with improved outcomes for patients [13], to such an extent that changes in proteinuria can be considered a surrogate endpoint for the progression of renal damage in randomized controlled trials (RCTs) [14]. It is now generally accepted that an albuminuria reduction of 30% from baseline, after the initiation of albuminuria-lowering therapy, confers a significant and strong protection against ESKD [15,16].

In recent years, targeting the renin-angiotensin-aldosterone system (RAAS) has been the only pharmacological approach for reducing proteinuria and improving renal outcomes in patients with CKD. In recent decades, novel sodium-glucose cotransporter 2 (SGLT2) inhibitors (SGLT2i) became the new pillar of treatment for people with type 2 diabetes mellitus (T2DM) and proteinuric CKD, since more and more studies demonstrated the nephroprotective and antiproteinuric effect of these drugs [17,18,19,20,21]. In fact, the inhibition of glucose reuptake in the proximal tubule allows for the restoration of the tubuloglomerular feedback mechanism, resetting glomerular haemodynamic, reducing hyperfiltration, and consequently, reducing proteinuria and the resulting renal damage.

While new trials have been conducted to assess the SGLT2i’s protective effects on cardiovascular and renal events in patients without T2DM and with normoalbuminuria, it has already been shown that many patients (up to 50% of patients) do not respond to these drugs in terms of proteinuria reduction [22]. Currently, the reason for the variability in proteinuria response in patients treated with SGLT2i is not entirely clear. Moreover, the ability to identify pre-treatment characteristics that can guide clinicians in therapy selection has yet to be established.

We conducted a retrospective study to evaluate clinical and laboratory variables that can help to predict proteinuria reduction with SGLT2i therapy in people with T2DM and CKD.

## 2. Materials and Methods

### 2.1. Study Design and Population

This observational retrospective, single-centre study was conducted at IRCCS Azienda Ospedaliero-Universitaria Policlinico Sant’Orsola (Bologna, Italy) by the Nephrology, Dialysis and Renal Transplantation Unit and the Endocrinology and Diabetes Prevention and Care Unit. All the patients included in this study were referred to our multidisciplinary outpatient service from 1 April 2019 to 9 February 2022.

The inclusion criteria were patients aged > 18 years, with a diagnosis of T2DM and CKD, and an expressed informed consent. All patients included in the study cohort completed a cycle of therapy with currently available SGLT2i (Dapaglifozin, Empagliflozin, Canagliflozin and Ertugliflozin). A cycle was defined as the period from the start of therapy at the first outpatient visit to the second outpatient visit during follow-up after therapy initiation. The length of follow-up was not predetermined but followed the dictates of clinical practice. Patients who discontinued therapy due to non-compliance or adverse events were not included in the study. Compliance with therapy was assessed via the presence of glucosuria on a urine dipstick.

The diagnosis and classification of CKD followed the 2012 Kidney Disease Improving Global Outcomes (KDIGO) guidelines [23]. Any patient with signs of renal impairment confirmed for at least 3 months, in terms of pathological proteinuria classified as at least A2 stage in the KDIGO classification (≥150 mg/24 h) and/or a reduction in the estimated glomerular filtration rate (eGFR) with values ranging between 90 mL/min/1.73 m^2^ and 30 mL/min/1.73 m^2^ (stage G2 and G3 in the KDIGO classification), was eligible for inclusion in the study. People with CKD stages G4 and G5, those on dialysis and kidney transplant recipients were excluded from the study cohort.

The cause of chronic kidney disease, although not diagnosed histologically, was likely attributable to diabetic kidney disease, and thus attributable to T2DM or to hypertensive and metabolic complications (dyslipidaemia, obesity). Patients with a histological diagnosis of glomerulonephritis or nephropathy secondary to immune, oncological or haematological diseases were excluded from the study. Similarly, patients with suspected glomerular pathology not secondary to diabetic disease or complications were excluded from the study. The study procedures were in accordance with the Helsinki Declaration. The study protocol and consent form were approved by the Ethics Committee of IRCCS Azienda Ospedaliero-Universitaria di Bologna (ID approval: 1008-2021).

From the first outpatient visit, data were retrospectively collected regarding medical history, current health status and ongoing therapy. In particular, we collected anamnestic data, including the presence of comorbidities such as cardiovascular disease and diabetic retinopathy, and anthropometric data, such as weight and height for calculating body mass index (BMI), and blood pressure measurements. Blood samples were also taken at the baseline visit to measure fasting blood glucose and glycated haemoglobin (Hba1c) levels, lipid panel (triglycerides, total cholesterol, low-density lipoprotein cholesterol (LDL), high-density lipoprotein cholesterol (HDL)) and measures of kidney function (serum creatinine and eGFR). Urine collection over 24 h was used to measure urinary excretion of proteins (uProt) expressed as mg/die (or mg/24 h). Finally, information regarding ongoing antidiabetic (metformin, insulin, DPP4 inhibitors (DPP4i), GLP1 receptor agonists (GLP1RA), thiazolidinedione, sulfonylureas), antihypertensive and nephroprotective therapy (ACE inhibitors (ACEi) and Angiotensin receptor blockers (ARB)) was collected.

From the second outpatient visit onwards, laboratory data were retrospectively collected, with a focus on the values of uProt, eGFR, BMI and HbA1c.

Proteinuria changes (both absolute and percentage) were calculated between baseline and post-treatment uProt levels. Patients were stratified into two subgroups based upon the response to the therapy with SGLT2i, namely the reduction in uProt of ≥30% from baseline levels. These two groups were called Responder (R) and non-Responder (nR).

### 2.2. Study Aim

The aim of the study was to analyse differences in baseline characteristics between the two groups and to investigate the relationship between them and the proteinuria changes. Additionally, relationships between proteinuria changes and variations in other laboratory variables, both from a renal and diabetological point of view, were explored.

### 2.3. Statistical Analysis

The statistical analysis was performed with R Software (version 2022.07.01+554). Descriptive statistics were used to analyse differences between R and nR groups at baseline and at follow-up. Categorical variables were expressed as percentage; continuous variables were expressed as means ± standard deviation (parametric variables) or median and interquartile range (IQR) (nonparametric variables). The Kruskal–Wallis test and unpaired t-test were employed to test the differences between the means of the two groups in both skewed and symmetric variables. The Chi-squared test was used to test the difference of percentage (%) between the two groups. Linear regressions were utilized to analyse the relationship between proteinuria change (absolute change) and baseline characteristics in both univariate and multivariate analyses. Basal characteristics were included in the multivariate analysis by selecting “a priori” the most known variables associated with proteinuria (uProt, eGFR, BMI, Hba1c). Linear regressions were also used to test the relationship between changes in proteinuria with eGFR and HbA1c variation, respectively. Finally, univariate and multivariate logistic regressions were conducted to analyse the association between baseline characteristics and changes in proteinuria among baseline and post-treatment visits. The sample size was determined based on the expected dimension, depending on the recruitment rate at the participating centre. A *p*-value of <0.05 was considered statistically significant.

## 3. Results

A total of 58 patients were enrolled in the study: 32 patients (55.1%) were in the R group and 26 patients (44.9%) in the nR group. Baseline characteristics of the cohort population and the two groups are summarized in Table 1. R patients were younger (64.1 ± 10.1 vs. 68.7 ± 10.6 years), had higher values of eGFR (61.4 ± 24.5 vs. 53.3 ± 18.5 mL/min/1.73 m^2^) and had a significantly higher uProt (1393 vs. 449 mg/24 h, *p* = 0.010). Thus, R patients had a higher prevalence of diabetic retinopathy but a lower prevalence of history of CVD (50% vs. 25% and 34.4% vs. 57.7%, respectively). There were no significant differences between the two groups in terms of BMI, blood pressure control, Hba1c levels or lipid panel. Approximately 90% of patients were in therapy with RAAS inhibitors titrated to the maximum tolerated dose, and there were no differences between the two groups. In terms of antidiabetic therapy, metformin was the most common drug followed by insulin, GLP1-RA and DPP4i. Each of these drugs was equally distributed between R and nR.

Dapagliflozin was the most common SGLT2i (60.3%), followed by Empagliflozin (19.0%), Canagliflozin (17.2%) and Ertugliflozin (3.4%), and there were no significant differences in the distribution of these molecules between R and nR.

The results of univariate and multivariate linear regressions assessing the correlation between baseline characteristics of all patients in the study cohort and the variation in proteinuria after the initiation of SGLT2 inhibitors are summarized in Table 2. There was a significant correlation between the uProt at baseline and proteinuria reduction with SGLT2i in both the univariate analysis (β = −0.43, CI −0.55 to −031; *p* < 0.001) and in the multivariate model (β = −0.46, CI −0.57 to −0.35, *p* < 0.001). The graphic correlation between uProt and proteinuria reduction is shown in Figure 1. There were no significant correlations with eGFR and BMI at baseline in the univariate linear regressions, but in the multivariate analysis, there was a significant positive correlation between eGFR and albuminuria reduction (β = −17, CI −31 to −3.3, *p* = 0.016) and a significant negative correlation with BMI (β = 81, CI 13 to 50, *p* = 0.021). No correlations with Hba1c levels and age at baseline were observed in either the univariate or the multivariate analyses.

The results of the univariate and multivariate logistic regressions assessing the correlation between baseline comorbidities (CVD and diabetic retinopathy) and the response to treatment in terms of a reduction in proteinuria of more than 30% are summarized in Table 3. In both the univariate and the multivariate models, diabetic retinopathy at baseline was associated with higher odds of being in the R group (Odds Ratio (OR) 2.76, CI 0.82 to 9.30, *p* = 0.09 in univariate; OR 3.65, CI 0.97 to 13.58, *p* = 0.054 in multivariate). On the other hand, the presence of CVD at baseline is associated with lower odds of being in the R group (OR 0.40, CI 0.13 to 1.17, *p* = 0.09 in univariate; OR 0.34, CI 0.09 to 1.22, *p* = 0.1 in multivariate). Neither of these statements reached statistical significance. The multivariate model is shown graphically in Figure 2.

Table 4 shows the BMI, eGFR and Hba1c values at follow-up, along with their percentage variation from baseline values. The mean follow-up time was 5.43 months with no significant difference between R and nR (5.45 vs. 5.40 months, *p* = 0.928). There were no significant differences in these variables between the two groups. However, we observed a significant difference (*p* = 0.017) in the ΔeGFR between the R and nR groups, with a decrease (−10.43%) in the R group and an increase (+2.31%) in the nR group. Additionally, we observed a trend toward statistical significance (*p* = 0.052) in the change of Hba1c values between the R and nR groups, with a decrease in the R group (−5.97%) and an increase in the nR group (+3.70%). Univariate linear regression analyses were used to test the correlation between changes in uProt and Hba1c, as well as between percentage changes in uProt and eGFR at follow-up (Figure 3). There was a positive association between ΔuProt and ΔeGFR, although this did not reach statistical significance (β = 0.05, CI 0.02 to 1.93, *p* = 0.059). However, there was a significant positive correlation between ΔuProt and ΔHba1c (β = 0.06, CI 0.02 to 2.85, *p* = 0.006).

## 4. Discussion

In recent years, several randomized controlled trials have investigated the nephroprotective effect of SGLT2i therapy, mainly in terms of risk reduction in renal-specific outcomes such as a reduction in eGFR, onset of end-stage kidney disease and death from renal causes [17,18,19]. Some studies have also demonstrated the effect of SGLT2i on reducing albuminuria [20,21,22,24]. The aim of this study was to investigate the antiproteinuric effect of SGLT2i therapy in people with T2DM and to identify factors that can predict the extent of the response to treatment in terms of proteinuria reduction.

In renal-specific RCTs such as DAPA-CKD and CREDENCE, the mean value of albuminuria, calculated as the urinary albumin-to-creatinine ratio (UACR), was 923 mg/g and 1017 mg/g, respectively, in people with T2DM receiving SGLT2i therapy. These studies demonstrated the antiproteinuric effect of these agents, with a mean reduction in albuminuria against a placebo of −31% (CI 26 to 35, *p* < 0.0001) in the CREDENCE trial and −35.1% (CI 39.4 to 30.6; *p* < 0.0001) in people with T2DM in the DAPA-CKD trial. Our study population is comparable, with a mean value of urinary protein (uProt) of 1170 mg/24 h [20,21].

The present study revealed that the baseline level of proteinuria was the most significant predictive factor of the response to treatment in terms of a reduction in proteinuria of more than 30% from baseline, with the R group having a higher baseline value of proteinuria. As mentioned above, the CREDENCE and DAPA-CKD trials demonstrated a reduction in proteinuria of approximately 30%. However, these trials only included patients with macroalbuminuria at baseline (UACR >300 mg/g in the CREDENCE trail and >200 mg/g in the DAPA-CKD trial) [20,21].

On the other hand, another study included patients with both micro and macroalbuminuria (UACR ranging from 30 to 3500 mg/g), with a mean UACR value of 270 mg/g in the dapagliflozin group, and a mean reduction in albuminuria relative to the placebo of −21.0% (CI −34.1 to −5.2; *p* = 0.011) [25]. In a post hoc analysis of the DAPA-CKD trial, it was shown that patients with a UACR >1000 mg/g had a greater response to treatment than those with a UACR <1000 mg/g (−31.7% vs. −26.9%), although this difference was not statistically significant [26]. Similarly, in a post hoc analysis of the EMPA-REG OUTCOME trial, patients were stratified based on their baseline albuminuria levels; it was observed that at 12 weeks, the mean reduction in albuminuria was directly proportional to the baseline levels of albuminuria: normoalbuminuric −7% (CI −12 to −2, *p* = 0.013), microalbuminuric −25% (CI −31 to −19, *p* < 0.0001) and macroalbuminuric −32% (CI −41 to −23, *p* < 0.0001) [27]. In a post hoc analysis of the CREDENCE trial, patients were divided into four groups based on their proteinuric response to treatment with SGLT2i (a reduction in UACR of 30% or more, reduction in UACR of between 0% and 30%, minor increase in UACR of between 0% and 30%, and more than 30% increase in UACR); this study showed that patients with a higher reduction in albuminuria had a higher UACR value at baseline (*p* < 0.001) [28]. As the current study has also demonstrated, there is a link between the value of proteinuria levels at baseline and proteinuria reduction after treatment with SGLT2i.

It is well known that proteinuria is a common feature of diabetic nephropathy and DKD. These conditions are characterized by a phenomenon called glomerular hyperfiltration, which involves a supraphysiologic elevation in eGFR that is mainly induced by alterations in glucidic metabolism, systemic hypertension, obesity and vascular and tubular factors in the kidney [29,30,31]. This phenomenon has been divided into two subgroups known as ‘whole kidney’ hyperfiltration, which is typical of the early stages of diabetic nephropathy, precedes the onset of albuminuria and can be accounted for by haemodynamic adjustment in the glomerulus, and ‘single nephron’ hyperfiltration, an adaptive response driven mainly by inflammatory and profibrotic molecules in the nephrons that characterizes the intermediate and advanced phases of DKD [32]. The main randomized controlled trials of SGLT2i therapy have shown an acute slope of eGFR relative to the placebo in the first weeks of treatment. The explanation for this phenomenon may be related to a reduction in glomerular hyperfiltration mediated by an adenosine-related mechanism on the afferent arteriole [33]. A study has also demonstrated the capacity of SGLT2i to reduce hyperfiltration in the first weeks of treatment in patients with T1DM and eGFR >135 mL/min/1.73 m^2^ calculated by PAI clearance [34]. The explanation for the finding that higher values of proteinuria at baseline are related to a better antiproteinuric response to treatment, and more specifically in patients with higher values of eGFR at baseline as shown in the multivariate model, may be related to the likelihood of the presence of ‘whole kidney glomerular hyperfiltration’, which is more susceptible to the acute haemodynamic effect of SGLT2i. Furthermore, patients who responded to treatment showed a reduction in eGFR at follow-up of approximately 10%, which provides further evidence for the haemodynamic effect of SGLT2i in the presence of glomerular hyperfiltration.

In this study, although statistical significance was not reached, a trend in terms of treatment response based on comorbidities was highlighted. In particular, diabetic retinopathy seems to be positively associated with a reduction in proteinuria, while CVD is negatively associated. DKD is characterized by a spectrum of phenotypes in which the most common one, the so-called ‘classical’ phenotype, is characterized by glomerular hyperfiltration, the onset and worsening of albuminuria and a subsequent decline in eGFR. Furthermore, a ‘non-proteinuric’ phenotype has been described as being characterized by the absence of macroalbuminuria but a progressive decline in eGFR [35]. A retrospective study observed that non-proteinuric people with T2DM had less typical lesions of diabetic nephropathy in renal biopsies compared to proteinuric patients [36]. Another study demonstrated that these lesions were observed with less frequency in the kidney biopsies of patients with normoalbuminuria than in patients with albuminuria and that the former had a higher rate of arteriosclerosis [37]. Some authors maintain that retinopathy precedes the onset of albuminuria in diabetic patients [35]. Thus, the absence of retinopathy and albuminuria may support a pathway of DKD progression that is distinct from the classical phenotype.

The results of the current study suggest that higher baseline proteinuria levels and the presence of retinopathy are associated with a greater reduction in proteinuria with SGLT2i treatment. Conversely, lower baseline proteinuria levels, the absence of retinopathy and the presence of CVD may indicate a non-proteinuric phenotype of DKD or other factors beyond DKD only, such as hypertension, atherosclerosis and obesity, where the antiproteinuric effect of SGLT2i may be less pronounced. Therefore, different clinical phenotypes of DKD may have different responses to SGLT2i treatment in terms of proteinuria reduction, with a stronger response in classical DKD. Further research is needed to confirm these findings and to investigate the potential impact of different DKD phenotypes on treatment response.

Jongs and colleagues demonstrated a significant correlation between Hba1c levels at baseline and the albuminuria reduction. Nonetheless, people with T2DM had a higher reduction in albuminuria compared to those without T2DM, suggesting a glucose mediated effect in SGLT2i therapy. In the current study, there was no significant correlation between Hba1c levels and the response to treatment. In their post hoc analysis, Jongs and colleagues divided patients into two groups based on Hba1c levels being below or over 8.5% [26]. In the current study, the mean value of Hba1c was 7.3%, and it was slightly higher in the Responder subgroup than in the non-Responder subgroup. Therefore, it could be concluded that, in this study, patients that were still treated with the standard of care for T2DM did not present a non-controlled T2DM in which Jongs observed a higher rate of response to treatment. Moreover, it is known that glomerular hyperfiltration recognizes a glucose-dependent pathway based upon the upregulation of SGLT2 receptors and a reduction in tubular-glomerular feedback, as well as a glucose-independent pathway mainly mediated by other molecules such as Angiotensin II, NO, Endothelin-1 and Trombaxan-A2 [33,38]. Therefore, the antiproteinuric effect of SGLT2i may recognize both glucose-dependent and -independent pathways. However, the current study found an association between a higher antiproteinuric response and a reduction in Hba1c. The meaning of these associations is not entirely clear, but one possible explanation is that better glucose control can help to reduce proteinuria with SGLT2i therapy. Alternatively, the enhanced response to treatment may be related to a higher degree of glucosuria and, therefore, a greater reduction in Hba1c in responsive patients. Very few studies have investigated the relationship between the degree of glucosuria and kidney outcomes, suggesting a better renal survival with higher values of glucosuria. Further research is needed to determine whether these variables can be useful tools for monitoring treatment responses [39,40].

Obesity is a common comorbidity in diabetic patients that can increasingly complicate kidney involvement in DKD. Nowadays, obesity and overweight are commonly acknowledged as risk factors not only for T2DM but also for CKD in people without T2DM [41]. Thus, obesity is responsible for kidney impairment due to a mechanism that is not related to T2DM and that is implied in both ‘whole kidney’ and ‘single nephron’ glomerular hyperfiltration through unique mechanisms such as perirenal fat deposition, adiponectin dysregulation, RAAS activation, aldosterone overexpression and mineralocorticoid receptor ligand-independent activation [32,42,43,44,45,46]. In the current study, in the multivariate regression model, BMI was found to be associated with a poorer response to treatment with SGLT2 inhibitors. The reason for this is not entirely clear, but it may be because SGLT2 inhibition in diabetic and obese patients cannot act on all pathways involved in glomerular hyperfiltration in these kinds of patients. More studies are needed to investigate the long-term effects of SGLT2i in these patients and to explore the pharmacological association, for instance, with GLP1RA, which has shown promising effects in kidney protection in people with T2DM and obesity, or mineralocorticoid receptor antagonists such as novel non-steroidal Finerenone.

One last point of discussion may be that SGLT2 inhibitors have a greater effect on reducing proteinuria in patients with higher baseline values. Recent studies have demonstrated consistent nephroprotective effects across all levels of albuminuria [22]. Furthermore, higher baseline proteinuria levels may indicate more severe and resistant glomerular disease. In a post hoc analysis of the CREDENCE trial, canagliflozin, an SGLT2 inhibitor, reduced albuminuria by 163, 355 and 341 mg/g in patients with baseline albuminuria levels of <1000, 1000–3000 and ≥3000 mg/g, respectively. However, the percentage change in albuminuria between the treatment and placebo groups was even higher in patients with lower baseline albuminuria levels [47]. Therefore, the results of the current study may indicate that higher levels of proteinuria alone cannot reliably indicate treatment response. Instead, they should be considered alongside other variables such as eGFR, BMI and comorbidities to identify a specific pattern of diabetic kidney disease phenotype that may be more responsive to SGLT2 inhibitor therapy in terms of proteinuria reduction.

## 5. Study Limits

The limitations of this study were the cohort population size and the retrospective analysis. Further studies are necessary which aim to investigate the relationship between baseline variables and proteinuria reduction with SGLT2i therapy.

## 6. Conclusions

In conclusion, in the current real-life experience, after the administration of SGLT2i therapy in DKD patients, a reduction of more than 30% in proteinuria was observed in more than half of the patients. However, a significant percentage of patients do not adequately respond to SGLT2i therapy in terms of reducing proteinuria. Baseline proteinuria seems to be a reliable characteristic that can help predict the response to treatment before therapy initiation. eGFR and BMI are variables that, in association with proteinuria, can help predict treatment response. Anamnestic features, such as cardiovascular disease and baseline proteinuria, could be considered in patient assessment before starting SGLT2i therapy.

Different phenotypes of DKD may have an impact in terms of antiproteinuric response. Patients with important antiproteinuric effects indeed presented increased levels of albuminuria at baseline and more frequently presented retinopathy. On the other hand, patients with lower baseline proteinuria and CVD presented a reduced antiproteinuric response to the drug. The long-term impact on kidney function of these different perspectives must be explored, but interestingly, a different pathogenetic pathway may present a different response to gliflozin.

## Figures and Tables

**Figure 1 life-13-01061-f001:**
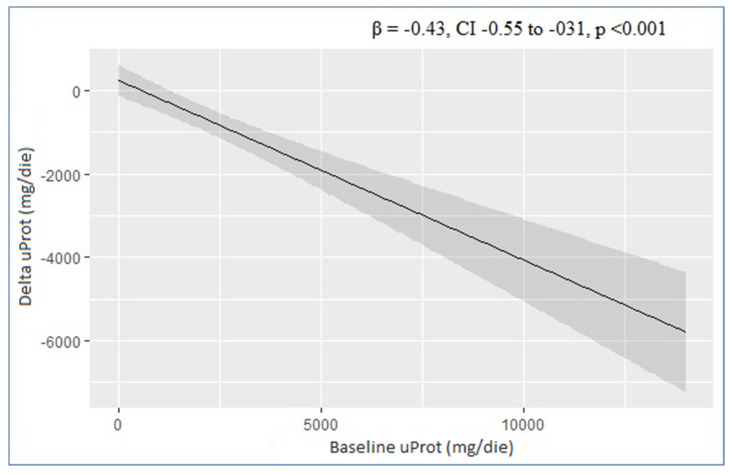
Linear regression analysis: correlation between baseline proteinuria and proteinuria variation (delta Δ). CI is represented in grey.

**Figure 2 life-13-01061-f002:**
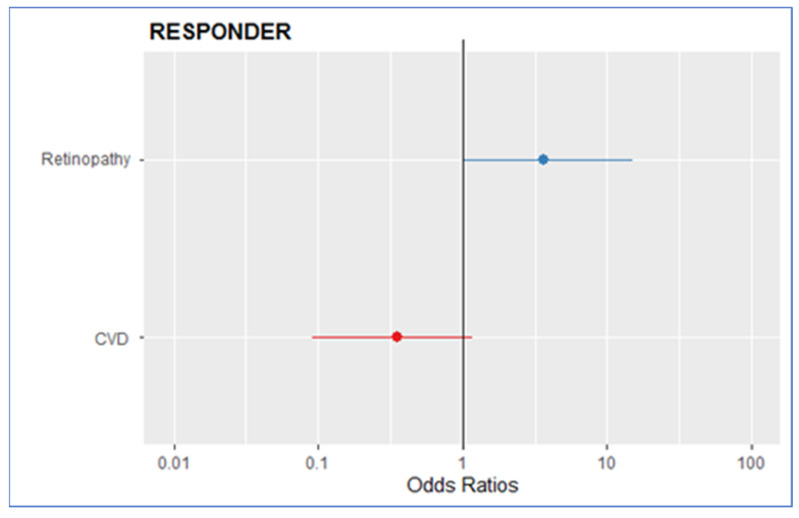
Multivariate logistic regression model: correlations between CVD and retinopathy with a reduction in uProt of >30%.

**Figure 3 life-13-01061-f003:**
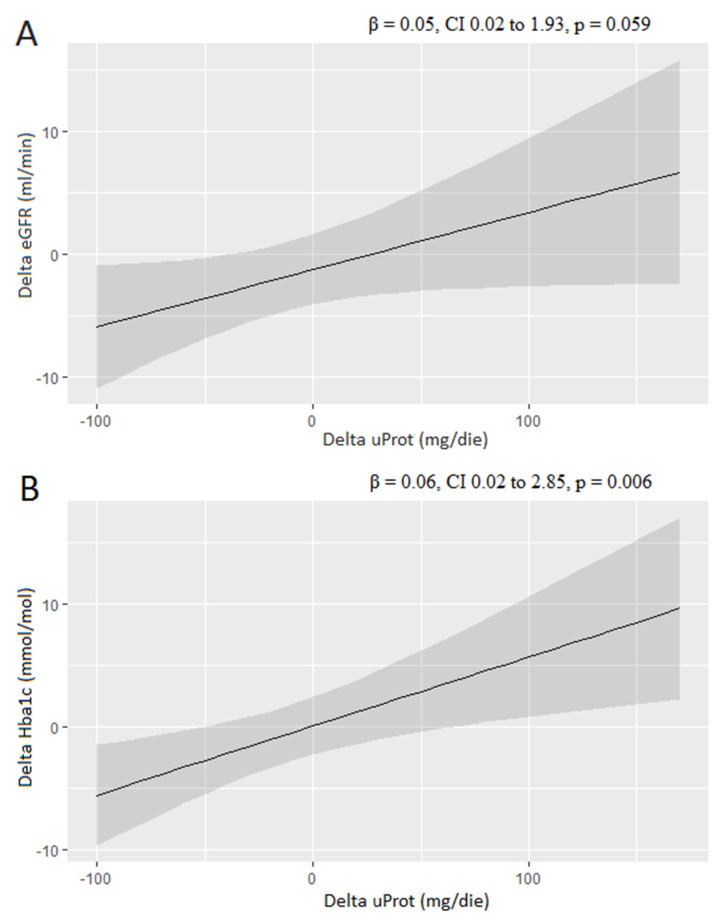
Linear regression analysis: correlation between ΔuProt and ΔeGFR (**A**) and between ΔuProt and ΔHba1c (**B**). CI is represented in grey.

**Table 1 life-13-01061-t001:** Basal characteristics of patients: overall and by proteinuria reduction groups.

	Overall (n = 58)	Non-Responder <30% (n = 26)	Responder ≥30% (n = 32)	*p*
Age years	66.1 ± 10.5	68.7 ± 10.6	64.1 ± 10.1	0.094
Male gender. *%*	77.6	84.6	71.8	0.247
Diabetes duration. years	15.0 [7.83–22.2]	12.1 [7.4–22.37]	15.5 [8.3–19.1]	0.580
BMI. Kg/m^2^	28.9 ± 4.4	29.1 ± 5.1	28.8 ± 3.8	0.843
CVD. *%*	44.8	57.7	34.4	0.076
Retinopathy. %	38.0	25.0	50.0	0.069
SBP. mmHg	130 ± 16	131 ± 16	130 ± 16	0.717
DBP. mmHg	79 ± 8	80 ± 9	79 ± 8	0.823
eGFR. mL/min/1.73 m^2^	57.8 ± 22.2	53.3 ± 18.5	61.4 ± 24.5	0.165
Glycated haemoglobin. mmol/mol	56.3 ± 11.0	55.2 ± 10.9	57.2 ± 11.1	0.506
Cholesterol. mg/dL	178 ± 49	173 ± 49	182 ± 49	0.543
LDL cholesterol. mg/dL	106 ± 37	105 ± 42	107 ± 33	0.841
Triglycerides. mg/dL	157 [113–208]	174 [109–219]	138 [120–201]	0.649
uProt. mg/24 h	1170 [360–2800]	449 [168–1680]	1393 [918–2920]	0.010
SGLT2i. n [%]				0.144
-Dapaglifozin	35 [60.3]	13 [37.2]	22 [62.8]	
-Empaglifozin	11 [19.0]	8 [72.7]	3 [27.3]	
-Canaglifozin	10 [17.2]	5 [50.0]	5 [50.0]	
-Ertuglifozin	2 [3.4]	0 [0.0]	2 [100.0]	
RAAS-inhibitors. % pts	89.7	88.5	90.6	0.788
Insulin, % pts	37.9	34.6	40.6	0.639
DPP4, % pts	19.0	19.2	18.8	0.963
GLP1RA, % pts	31.0	34.6	28.1	0.595
Methformin, % pts	55.2	61.5	50.0	0.380
BP-lowering drugs, n	2.1 ± 1.1	2.2 ± 1.1	2.1 ± 1.1	0.750

**Table 2 life-13-01061-t002:** Univariate and multivariate linear regression models: correlations between proteinuria changes and basal characteristics in both univariate and multivariate linear regressions.

	Univariate			Multivariate		
Characteristic	Beta	95% CI	*p*-Value	Beta	95% CI	*p*-Value
uProt (mg/24 h)	−0.43	−0.55, −0.31	<0.001	−0.46	−0.57, −0.35	<0.001
eGFR (mL/min/1.73 m^2^)	−10	−29, 8.5	0.3	−17	−31, −3.3	0.016
BMI (kg/m^2^)	50	−49, 149	0.3	81	13, 150	0.021
Hba1c (mmol/mol)	19	−19, 58	0.3	13	−15, 40	0.4
Age (years)	−3.6	−44, 37	0.9	−14	−44, 15	0.3

**Table 3 life-13-01061-t003:** Univariate and multivariate logistic regression models: correlations between a reduction in uProt of >30% and comorbidities (CVD, retinopathy).

	Univariate			Multivariate		
Characteristic	OR	95% CI	*p*-Value	OR	95% CI	*p*-Value
CVD	0.40	0.13, 1.17	0.09	0.34	0.09, 1.22	0.1
Retinopathy	2.76	0.82, 9.30	0.09	3.65	0.97, 13.58	0.054

**Table 4 life-13-01061-t004:** Patients’ characteristics at follow-up.

	Overall (n = 58)	Non-Responder <30% (n = 26)	Responder ≥30% (n = 32)	*p*
BMI. Kg/m^2^	29.56 ± 6.56	28.10 ± 4.63	30.73 ± 7.65	0.104
Hba1c. mmol/mol	55.80 ± 11.41	57.54 ± 12.64	54.30 ± 10.21	0.554
eGFR. mL/min	55.84 ± 21.38	55.09 ± 19.35	56.47 ± 23.24	0.743
ΔBMI. %	−1.83 [−3.89, 0.53]	−0.64 [−3.54, 0.61]	−2.16 [−4.02, 0.36]	0.782
ΔHba1c. %	−1.72 [−10.50, 11.11]	3.70 [−6.39, 13.11]	−5.97 [−10.87, 4.88]	0.052
ΔeGFR. %	−2.55 [−11.92, 7.89]	2.31 [−5.93, 8.05]	−10.43 [−16.59, 2.54]	0.017

## Data Availability

The data presented in this study are available on request from the corresponding author. The data are not publicly available available due to privacy concern.

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
