# Peer review of "Impact of Baseline Clinical Variables on SGLT2i’s Antiproteinuric Effect in Diabetic Kidney Disease"

_life, 2023, doi:10.3390/life13041061_

Round 1
Reviewer 1 Report
Comments to the Authors,
Minor comments:
· It is preferable to use the term people with type 2 diabetes than type 2 diabetes patients all through the text.
· It is preferable to avoid using the terms we and our and replace them with the current study.
Major comments:
· The manuscript requires thorough English revision because of spelling and linguistic errors.
· Title: The title is vague. It’s preferred to use a more precise title.
· Methodology:
· Was ethical committee approval taken and if so could you please add its number.
· The study was stated to be retrospective study, however, it’s written that patients were studied at baseline and at the study endpoint after five months. Isn’t this a prospective study please verify the type of study.
· Was sample size calculated to detect if the sample size is appropriate?
· It is preferable to add data about the compliance of the patients and their concurrent medications.
· Results:
· It is preferable to add data about the percent change of each of the variables at the study endpoint from the baseline data.
· It would be nice to add correlations for factors associated with proteinurea percent changes.
· As shown in table 1, people with T2DM who were non-responders didn’t show significant decrease in their protein level; however, this could be because there proteinurea level was not high enough.

Author Response
Dear Nouran Y. Salah,
Thank you for the opportunity to submit a revised version of our paper (Manuscript ID: life-2254834). Below we answer the reviewers’ criticisms point by point. Our replies and changes to the text are in boldface or reported as ‘track-changes’ to allow a quick check. We appreciate the time and effort of the reviewers for their thoughtful feedback. We feel that the revised version of this manuscript has benefited considerably from the review process. We hope that we have addressed each of the issues raised adequately. Please feel free to contact us with any additional questions or comments.
We look forward to your final decision on this paper.
Yours sincerely
Gaetano La Manna
Minor comments:
- It is preferable to use the term people with type 2 diabetes than type 2 diabetes patients all through the text.
- It is preferable to avoid using the terms we and our and replace them with the current study.
We have modified the lexicon and terminology in the article as suggested.
Major comments:
- The manuscript requires thorough English revision because of spelling and linguistic errors.
A linguistic check was performed and the detected errors were corrected.
- Title: The title is vague. It’s preferred to use a more precise title.
We changed it as requested.
Methodology:
- Was ethical committee approval taken and if so could you please add its number.
Yes, it has been approved by the ethics committee. We added it in the methodology section.
- The study was stated to be retrospective study, however, it’s written that patients were studied at baseline and at the study endpoint after five months. Isn’t this a prospective study please verify the type of study.
The study is retrospective in nature, as the data were retrospectively collected from outpatient evaluations at our clinic in a previous historical moment with respect to the study idea. We do agree that, as it was written in the Materials and Methods section, it could have appeared to be prospective in nature. Therefore, we have modified that part to make it more understandable.
- Was sample size calculated to detect if the sample size is appropriate?
Sample size was evaluated on the basis of the expected dimension, depending on the recruitment rate of the participating center. We have now added a sentence in the Methods section.
- It is preferable to add data about the compliance of the patients and their concurrent medications.
We added in the methodology the concurrent antidiabetic, antihypertensive and nephroprotective medications. We retrospectively selected patients that were compliant to SGLT2i. Patients who discontinued the medication due to poor compliance or adverse events were excluded. We make it more clear in the methodology as you suggested.
Results:
- It is preferable to add data about the percent change of each of the variables at the study endpoint from the baseline data.
This was very helpful advice, so we followed it and conducted a linear regression analysis to investigate the correlation between the percent change in the selected variables and proteinuria reduction. We also included comments on these findings in both the Results and Discussion sections.
It would be nice to add correlations for factors associated with proteinuria percent changes.
We added it.
- As shown in table 1, people with T2DM who were non-responders didn’t show significant decrease in their protein level; however, this could be because there proteinurea level was not high enough.
This is a very interesting comment. From one side, we knew from recent phase 3 trials (e.g. doi: 10.1056/NEJMoa2204233, the EMPA-KIDNEY study) that SGLT2 inhibitors have positive effects also in patients with low-moderate proteinuria. But, with this specific aim, we compared our analysis with a post-hoc analysis of the CREDENCE trial (doi: 10.2215/CJN.15260920). In this analysis, Authors found that the SGLT2i canagliflozin decreased albuminuria by 163, 355 and 341 mg/g in patients with baseline albuminuria <1000, 1000-3000 and ≥3000 mg/g, respectively but the % change of albuminuria between treatment and placebo group was even higher in patients with lower baseline albuminuria (testifying a greater treatment effect in low Proteinuric group). However, this was a good point to reflect, hence we added few sentences in discussion.

Reviewer 2 Report
Thank you for providing an interesting article. Cepelli et al present a retrospective analysis of 58 patients with T2DM and CKD who were commenced on SGLT-2 inhibitors and analyzed the patients in terms of reduction in proteinuria over the follow up period.
The introduction is well described with appropriate references setting the background. In addition, the discussion is thorough.
Comments:
1. Line 40-41: “eGFR and BMI are reliable characteristics that can help predict the response to treatment before therapy initiation” – consider toning down this statement. This is retrospective study with small patient numbers, there were no significant differences between BMI and eGFR for the R vs nR groups as well as no relationship captured on univariate regression. Only on multivariate was it captured. See comment #11 as well.
2. Line 103: For inclusion criteria, can you please clarify whether patients on dialysis or had kidney transplantation were included in this study?
3. Line 106: Do you have any data on the cause of CKD for the patients? Do all of the patients have diabetic kidney disease? Any other causes of CKD included? What is the duration of T2DM prior to inclusion in the study?
4. Line 129: You have indicated that mean follow-up was 5 months. Can you also please specify the follow up period separately for the R and nR groups – any significant differences? Have you considered if you need to adjust for follow-up time in the regression analysis? How does the 5 months follow up compare to previous studies which examined the proteinuria effect of SGLT-2 inhibitors?
5. Line 134: It is inappropriate to call them both "control" groups
6. Line 164: For Figure 1 please specify in the legend if this univariate or multivariate regression and please indicate what the grey shading around the regression line indicates (?CI). Also please add the Beta to the graph next to the CI range.
7. Line 171: To clarify, is this regression model based on the total cohort 58 patients? Y = reduction in proteinuria, X=baseline proteinuria for all 58 patients?
8. Line 174 and 185: please specify which variables were included in the multivariate regression models.
9. Line 191: Do you have results of univariate logistic regression for CVD and retinopathy? I can only see a multivariate regression reported and it is unclear which variables were included in this multivariate regression model.
10. Lines 257-259: You detail diabetic retinopathy and CVD as factors of association in this section however in the results section both of these factors had confidence intervals that crossed 1. Please comment on this.
11. Line 304: You mention “BMI was a negative predictive factor of proteinuria reduction with SGLT2i” – this relationship was not evident in the univariate analysis but only in the multivariate (after adjusting for which variables?) – can you comment on the significance of this?
12. Line 313: Have you considered possibility of confounding for the associations you have described?
Author Response
Thank you for the opportunity to submit a revised version of our paper (Manuscript ID: life-2254834). Below we answer the reviewers’ criticisms point by point. Our replies and changes to the text are in boldface or reported as ‘track-changes’ to allow a quick check. We appreciate the time and effort of the reviewers for their thoughtful feedback. We feel that the revised version of this manuscript has benefited considerably from the review process. We hope that we have addressed each of the issues raised adequately. Please feel free to contact us with any additional questions or comments.
We look forward to your final decision on this paper.
Yours sincerely
Gaetano La Manna
- Line 40-41: “eGFR and BMI are reliable characteristics that can help predict the response to treatment before therapy initiation” – consider toning down this statement. This is retrospective study with small patient numbers, there were no significant differences between BMI and eGFR for the R vs nR groups as well as no relationship captured on univariate regression. Only on multivariate was it captured. See comment #11 as well.
We fully agree with this consideration and therefore we have rewritten the concept as suggested.
- Line 103: For inclusion criteria, can you please clarify whether patients on dialysis or had kidney transplantation were included in this study?
Patients on dialysis and kidney transplant recipients were not included in the study cohort. This information was not clear in the inclusion criteria, which we have therefore modified.
- Line 106: Do you have any data on the cause of CKD for the patients? Do all of the patients have diabetic kidney disease? Any other causes of CKD included? What is the duration of T2DM prior to inclusion in the study?
Doubts regarding the cause of CKD in the included patients are of significant relevance, so we have added a paragraph in the Materials and Methods section to address it. Regarding the duration of diabetic disease, we have included this information in the Results section and in Table 1.
- 1Line 129: You have indicated that mean follow-up was 5 months. Can you also please specify the follow up period separately for the R and nR groups – any significant differences? Have you considered if you need to adjust for follow-up time in the regression analysis? How does the 5 months follow up compare to previous studies which examined the proteinuria effect of SGLT-2 inhibitors?
We added the follow-up time of both groups and they were completely overlapping. Therefore, we do not believe it is necessary to stratify the analyses for follow-up. As for the RCTs, the decrease in proteinuria occurred acutely in the first month and then stabilized during the treatment. Therefore, we believe that values at 5 months are almost entirely overlapping with those at the end of the trials
- Line 134: It is inappropriate to call them both "control" groups
We have corrected the error.
- Line 164: For Figure 1 please specify in the legend if this univariate or multivariate regression and please indicate what the grey shading around the regression line indicates (?CI). Also please add the Beta to the graph next to the CI range.
We added this information a suggested.
- Line 171: To clarify, is this regression model based on the total cohort 58 patients? Y = reduction in proteinuria, X=baseline proteinuria for all 58 patients?
Yes, we specified it the text more clearly.
- Line 174 and 185: please specify which variables were included in the multivariate regression models.
We have added a paragraph to specify the regressions model, adding the univariate regressions as asked in point 9.
- Line 191: Do you have results of univariate logistic regression for CVD and retinopathy? I can only see a multivariate regression reported and it is unclear which variables were included in this multivariate regression model.
We added the univariate regression model.
- Lines 257-259: You detail diabetic retinopathy and CVD as factors of association in this section however in the results section both of these factors had confidence intervals that crossed 1. Please comment on this.
You are absolutely right; in fact, we have toned down that statement a bit. In our study, we found a trend in terms of positive response for retinopathy and negative response for CVC. Both did not reach statistical significance, but we believe it is due to the sample size. In particular, for retinopathy, which almost reached significance in the multivariate analysis, it is likely that with a larger sample size, significant results could have been obtained.
- Line 304: You mention “BMI was a negative predictive factor of proteinuria reduction with SGLT2i” – this relationship was not evident in the univariate analysis but only in the multivariate (after adjusting for which variables?) – can you comment on the significance of this?
We have toned down this statement in the Discussion section.
- Line 313: Have you considered possibility of confounding for the associations you have described
Yes, possible confounding factors could be the presence of non diagnosticated glomerulopathies in the nR subgroup.

Round 2
Reviewer 1 Report
No further comments.
Reviewer 2 Report
Thank you for addressing suggestions point by point.